# Serological Response Patterns to Assess Treatment Outcomes in Advanced Non-Small Cell Lung Cancer: A Real-World Exploratory Multi-Center Observational Cohort Study

**DOI:** 10.3390/cancers17223647

**Published:** 2025-11-13

**Authors:** Alessandra I. G. Buma, Femke Laarakker, Frederik A. van Delft, Milou M. F. Schuurbiers, Jasper Smit, Antonius E. van Herwaarden, Huub H. van Rossum, Michel M. van den Heuvel

**Affiliations:** 1Department of Respiratory Medicine, Radboud University Medical Centre, 6525 GA Nijmegen, The Netherlands; femke.laarakker@radboudumc.nl (F.L.); milou.schuurbiers@envida.nl (M.M.F.S.); m.m.vandenheuvel-22@umcutrecht.nl (M.M.v.d.H.); 2Department of Laboratory Medicine, Netherlands Cancer Institute, 1066 CX Amsterdam, The Netherlands; f.v.delft@nki.nl (F.A.v.D.); h.v.rossum@nki.nl (H.H.v.R.); 3Health Technology and Services Research Department, Technical Medical Centre, University of Twente, 7522 NB Enschede, The Netherlands; 4Department of Thoracic Oncology, Netherlands Cancer Institute, 1066 CX Amsterdam, The Netherlands; ja.smit@nki.nl; 5Department of Laboratory Medicine, Radboud University Medical Centre, 6525 GA Nijmegen, The Netherlands; teun.vanherwaarden@radboudumc.nl; 6Department of Respiratory Medicine, University Medical Centre Utrecht, 3584 CX Utrecht, The Netherlands

**Keywords:** serum tumor markers, prognostic biomarkers, response classification, clinical decision-making, immune checkpoint inhibitors, non-small cell lung cancer

## Abstract

Previous studies mainly investigated singular serum tumor marker (STM) measurements for the management of advanced cancer patients, resulting in differences between recommended cut-off points and associated accuracies in evaluating treatment outcomes. Our exploratory multi-center observational cohort study is the first to show that monitoring of STM dynamics during treatment can help to identify advanced non-small cell lung cancer (NSCLC) patients who are more or less likely to have long-term favorable treatment outcomes upon receiving immune checkpoint inhibitor (ICI)-containing treatment. This way, response classification and decision-making in clinical practice can be improved without the need to use specific cut-off points besides the measured STM’s upper reference limit. Importantly, the dynamics identified in our study can potentially also be applied for other tumor- and systemic treatment-types and other tumor cell analytes facing similar challenges to improve response classification and decision-making across multiple indications.

## 1. Introduction

The prognosis of non-small cell lung cancer (NSCLC) patients has historically been poor, with an overall survival (OS) of less than 50% within one year after diagnosis [1]. Fortunately, the introduction of immune checkpoint inhibitors (ICIs) has markedly improved survival rates in patients with advanced-stage disease, with durable responses being seen in approximately 20–30% of patients five years after treatment start [2]. A significant proportion of patients, however, experience clinical and/or radiological progression after a primary response due to acquired treatment resistance [3]. Furthermore, some patients may benefit from additional local treatment and continuing treatment even when radiological progression has been confirmed, while others maintain long-term disease control despite early treatment discontinuation [3,4,5]. Assessing treatment outcomes promptly in real-time would therefore be highly advantageous for decision-making in clinical practice.

Serum tumor markers (STMs) are circulating protein-based molecules that are released by tumor cells or other cells in the body in response to tumor growth [6]. Next to being inexpensive, rapidly determined, and the collection of required blood minimally invasive, their serum levels have shown to reflect tumor mass, making them valuable for evaluating a patient’s tumor status in real-time [7,8]. To date, studies investigating STMs for the management of advanced cancer patients predominantly focused on predicting prognosis and/or (early) treatment response based on singular STM concentrations measured before and/or shortly after treatment start, resulting in differences between recommended cut-off points and associated accuracies in evaluating treatment outcomes [9,10,11,12,13,14,15,16,17,18,19,20,21,22,23,24]. Which STM dynamics can be observed during systemic treatment in patients with disease control early after treatment start and whether these dynamics can be used to assess treatment outcomes, however, is yet unknown. As STMs directly derive from the tumor, the dynamics are expected to be comparable to other tumor cell analytes such as circulating tumor DNA (ctDNA).

In this proof-of-concept study, we aimed to (i) determine which STM dynamics recur during treatment in advanced NSCLC patients with disease control three months after start with ICI-containing treatment, (ii) subclassify these dynamics into three serological response patterns (e.g., serological remission (SeR), serological stable/unknown significance (SeS), and serological progression (SeP)), and (iii) explore the prognostic value of these patterns in terms of progression-free survival (PFS) and OS.

## 2. Materials and Methods

### 2.1. Study Design and Patient Population

We performed a real-world exploratory multi-center observational cohort study that included adults with advanced NSCLC who received ICI-containing treatment at the Radboud University Medical Centre (Radboudumc), Nijmegen, The Netherlands, or the Netherlands Cancer Institute (NKI), Amsterdam, The Netherlands, between March 2013 and January 2023. ICI-containing treatment was administered in accordance with corresponding study protocols or local guidelines that were applicable at treatment start (e.g., early access, compassionate use program, clinical trials, routine care). Patients were included if they had clinical and radiological disease control three months after treatment started, and if at least three STM measurements for at least one STM had been performed during treatment. Additional details regarding the inclusion criteria are provided in the Appendix A. Clinical disease control was defined as no deterioration or an improvement in clinical symptoms as assessed by the treating thoracic oncologist [25]. Radiological disease control was defined as stable disease (SD), partial response (PR), or complete remission (CR) based on tumor evaluation by the radiologist on Computed Tomography (CT) imaging according to Response Evaluation Criteria in Solid Tumors (RECIST) version 1.1 [26]. The study was performed in accordance with the Declaration of Helsinki and followed the Strengthening the Reporting of Observational Studies in Epidemiology (STROBE) reporting guideline for cohort studies [27]. The study was approved by the local ethics committee (Commissie Mensgebonden Onderzoek) of the Radboudumc and the Institutional Review Board of the NKI. All patients provided written informed consent.

### 2.2. Procedures

Clinical and radiological response assessments during treatment were performed by the treating thoracic oncologists and radiologists of the participating centers based on symptoms reported by the patient at each follow-up visit and tumor dynamics evaluated with RECIST version 1.1 on routinely performed CT imaging, respectively [25,26]. Patients with no clinical and/or radiological progression between three months and two years after treatment start were classified as having achieved a durable response, while patients who developed clinical and/or radiological progression between three months and two years after treatment start were classified as having developed secondary treatment resistance [26,28]. Baseline characteristics (including clinical data, medical history, smoking status, pack years, albumin level, kidney function, medication use, and tumor characteristics) were extracted from electronic patient records.

#### STM Analysis

STM data were retrospectively retrieved for all patients, collecting all routinely available measurements of cytokeratin 19 fragment antigen (Cyfra 21.1), carcinoembryonic antigen (CEA), and cancer antigen-125 (CA-125). Only measurements performed between baseline and seven days after the maximum treatment period of two years (in patients achieving a durable response) or the date of progression (in patients who developed secondary treatment resistance) were used for analysis. All STM levels were measured using a Cobas 6000, 8000, or Pro system (Roche diagnostics, Basel, Switzerland) according to the manufacturer’s instructions. The applied reference ranges were 0.0–1.9 μg/L for Cyfra 21.1, 0.0–5.0 μg/L for CEA, and 0.0–35.0 U/mL for CA-125. Haemolyzed samples were excluded from the analysis.

### 2.3. Statistical Analysis

Statistical analyses were performed using R (Version 4.4.2) and SPSS (Version 29). A two-sided *p*-value < 0.05 was considered statistically significant [29]. No statistical methods were used to predetermine sample size.

#### 2.3.1. Patient Characteristics

Patient characteristics at baseline were described for all patients. Continuous variables were reported as means with associated standard deviation (StD) or medians with associated interquartile range (IQR) for normally and non-normally distributed data, respectively. Categorical variables were reported as ratios. Intergroup comparisons were performed using one-way ANOVA, the Kruskal–Wallis test, or Fisher’s exact test.

#### 2.3.2. STM Dynamics During Treatment and Subclassification into Serological Response Patterns

All retrieved STM data were first used to visualize STM levels for each patient to identify dynamics recurring during treatment based on observation by two investigators (AB and FL) who were blinded for treatment outcomes. The recurring dynamics were then subclassified into three serological response patterns by the two investigators based on the STM concentration over time with respect to the STM’s upper reference limit and the height of the STM level itself within each patient, showing persistently (a) normalized STM levels (subclassified as SeR), (b) stable STM levels or STM dynamics of unknown significance (subclassified as SeS), or (c) increasing STM levels (subclassified as SeP). As a result, a classification for each patient was obtained for each available individual STM and all available STMs combined, assuming the following importance of the serological response patterns for treatment outcome when assessing the combined serological response classification: SeR < SeS < SeP. An overview of all possible individual serological response classifications and their corresponding combined serological response classifications is shown in Appendix A. Note that if not all STMs were measured within a patient, the combined serological response classification was based on the available individual serological response classifications only.

Sub-analyses were additionally performed for both the individual and combined serological response classifications to assess the added value of each serological response pattern in distinguishing (i) patients achieving a durable response versus patients who developed secondary treatment resistance, (ii) patients with secondary treatment resistance who developed oligoprogression versus systemic progression, (iii) patients achieving a durable response who received treatment for two years versus those who received treatment for less than two years, (iv) patients achieving a durable response who progressed after two years versus those who did not, and (v) patients in whom treatment was discontinued due to immune-related adverse events (irAEs) versus those in whom this was not [30]. Intergroup comparisons were performed using Fisher’s exact test.

#### 2.3.3. Prognostic Value Serological Response Patterns

Kaplan–Meier survival analyses were performed for the individual serological response classifications, the combined serological response classifications, and the STM dynamics subclassified as SeS, to explore the prognostic value of the serological response patterns and potential survival differences between the STM dynamics subclassified as SeS in terms of PFS and OS. Statistical significance was evaluated using the log-rank test. In addition, multivariable Cox regression analyses were performed for the individual serological response classifications and the combined serological response classifications, including all patient characteristics that were described at baseline as variables in the analyses. PFS was calculated from the treatment start date to the date of clinical and/or radiological progression, death, or last follow-up. OS was calculated from the treatment start date to the date of death or last follow-up. Median follow-up was calculated from the treatment start date to the date of death or last follow-up. All follow-up data were collected until 3 July 2024.

## 3. Results

### 3.1. Patient Characteristics

STM data of 256 patients were available for analysis, with a mean of 9.9 (StD: 6.1), 9.7 (StD: 6.1), and 8.7 (StD: 5.8) measurements for each patient for Cyfra 21.1, CEA, and CA-125, respectively (Figure 1).

Median follow-up was 19.1 months (interquartile range (IQR): 11.2–41.0), with 218 (*n* = 218/256; 85.2%) patients developing secondary treatment resistance and 200 (*n* = 200/256; 78.1%) deaths. At treatment start, patients had a median age of 64.0 (IQR: 56.0–71.0) years and the majority were female (Table 1). Elevated baseline levels of Cyfra 21.1, CEA, and CA-125 were present in 84.8% (*n* = 167/197), 62.0% (*n* = 119/192), and 59.5% (*n* = 103/173) of patients for whom a baseline measurement was available, respectively. Programmed death-ligand 1 (PD-L1) expression was negative or weak positive in most patients, resulting in more patients being treated with a combination of ICIs and chemotherapy than with ICI monotherapy. Furthermore, treatment was predominantly administered in the first-line setting. Patient characteristics across the serological response patterns for the individual and combined serological response classifications are summarized in Appendix A.

### 3.2. STM Dynamics During Treatment and Subclassification into Serological Response Patterns

STM levels visualized for all included patients revealed 12 distinct recurring dynamics, of which two were subclassified as SeR, six as SeS, and four as SeP (Figure 2). Notably, most patients had SeS for each STM (Cyfra 21.1: *n* = 117/256, 45.7%; CEA: *n* = 127/252, 50.4%; CA-125: *n* = 118/239, 49.4%) (Appendix A). Combined assessment of the individual serological response classifications resulted in 6 (*n* = 6/256, 2.3%), 89 (*n* = 89/256, 34.8%), and 161 (*n* = 161/256, 62.9%) patients having SeR, SeS, or SeP, respectively (Appendix A). Examples of STM levels visualized for three of the included patients and their allocated individual and serological response classifications are shown in Figure 3.

Sub-analyses showed added value of the serological response patterns in distinguishing (i) patients who achieved a durable response versus patients who developed secondary treatment resistance (Cyfra 21.1, *p* < 0.001; CEA, *p* = 0.007; CA-125, *p* = 0.008; combined serological response classification, *p* < 0.001), and (ii) patients with secondary treatment resistance who developed oligoprogression versus systemic progression (Cyfra 21.1, *p* < 0.001; CA-125, *p* = 0.022; combined serological response classification, *p* < 0.001) (Appendix A). No added value of the serological response patterns was found in any other clinical scenario (Appendix A).

### 3.3. Prognostic Value Serological Response Patterns

An overview of the median PFS and OS associated with each serological response pattern for the individual and combined serological response classifications and results obtained with the log-rank test can be found in Appendix A.

For Cyfra 21.1, median PFS and OS were significantly longer in patients with SeR (9.0 months (95% confidence interval (CI): 1.5–16.6); 35.0 months (95% CI: 28.9–41.1)) or SeS (9.0 months (95% CI: 7.0–10.9); 28.2 months (95% CI: 14.8–41.6)) compared to patients with SeP (6.4 months (95% CI: 5.7–7.1); 14.5 months (95% CI: 12.4–16.6)), respectively (Appendix A).

For CEA, median PFS and OS were significantly longer in patients with SeR (10.8 months (95% CI: 7.2–14.4); 91.1 months (95% CI: 5.6–176.6)) compared to patients with SeS (7.6 months (95% CI: 6.5–8.6); 19.6 (95% CI: 16.1–23.2)) or SeP (6.9 months (95% CI: 5.8–7.9); 17.7 months (95% CI: 14.8–20.6)), respectively (Appendix A).

For CA-125, median PFS and OS were significantly longer in patients with SeR (9.0 months (95% CI: 7.8–10.1); 25.5 months (95% CI: 16.2–34.7)) or SeS (7.2 months (95% CI: 6.5–8.0); 21.8 months (95% CI: 17.4–26.2)) compared to patients with SeP (6.8 months (95% CI: 5.5–8.2); 13.6 months (95% CI: 11.0–16.1)), respectively (Appendix A).

For the combined serological response classification, median PFS and OS were significantly longer in patients with SeR (10.8 months (95% CI: 0.0–43.7); not reached (NR) (95% CI: NR-NR)) or SeS (8.3 months (95% CI: 6.5–10.1); 29.7 months (95% CI: 15.5–43.9)) compared to patients with SeP (6.9 months (95% CI: 6.1–7.8); 16.5 months (95% CI: 14.0–18.9)), respectively (Figure 4 and Figure 5).

Results obtained for the STM dynamics subclassified as SeS are included in Appendix A.

Results obtained with multivariable Cox regression analysis can be found in Appendix A. Notably, the individual serological response classification for Cyfra 21.1 (hazard ratio (HR): 1.982 (95% CI: 1.356–2.897), *p* < 0.001; HR: 2.654 (95% CI: 1.823–3.863), *p* < 0.001) and the combined serological response classification (HR: 1.563 (95% CI: 1.039–2.353), *p* = 0.03; HR: 2.726 (95% CI: 1.682–4.418), *p* < 0.001) remained significant predictors for both PFS and OS, respectively. The individual serological response classifications for CEA (HR 1.826 (95% CI: 1.290–2.583), *p* < 0.001) and CA-125 (HR 1.774 (95% CI: 1.213–2.595), *p* = 0.003) remained significant predictors for OS only.

## 4. Discussion

This real-world exploratory multi-center observational cohort study represents the first comprehensive evaluation of STM dynamics observed during systemic treatment in advanced cancer patients with disease control early after treatment start. Importantly, we identified 12 distinct recurring dynamics in advanced NSCLC patients who received ICI-containing treatment, which could be grouped into three serological response patterns that were significantly associated with both PFS and OS. Notably, the serological response patterns could specifically be used to distinguish patients who achieved a durable response versus patients who developed secondary treatment resistance, and patients with secondary treatment resistance who developed oligo progression versus systemic progression. Monitoring of STM dynamics during treatment can help to assess treatment outcomes promptly, thereby improving response classification and decision-making in clinical practice.

Previous studies investigating the use of STMs in the management of cancer patients mainly dichotomized STM levels based on chosen cut-off points, resulting in differences between recommended cut-off points and associated accuracies in evaluating treatment outcomes [9,10,11,12,13,14,15,16,17,18,19,20,21,22,23,24]. In the present study, we demonstrate that treating STM levels as continuous variables and integrating repeated measurements retains important information on disease control without the need to apply specific cut-off points besides the measured STM’s upper reference limit. Specifically, we found a significant association between the serological response patterns and patients who achieved a durable response versus those who developed secondary treatment resistance, highlighting the potential of STM dynamics for predicting long-term treatment outcomes.

Various pathophysiological and technical factors are known to increase STM concentrations, leading to potential misinterpretation and unnecessary psychological stress for the patient due to the assumption of disease progression [31]. Interestingly, we observed that survival outcomes for Cyfra 21.1 and CA-125 did not significantly differ between patients with SeR and SeS, whereas for CEA patients with SeS had survival outcomes more similar to patients with SeP. These observations could be explained by the lower intra-individual biological variability and higher inter-individual biological variability of CEA compared to Cyfra 21.1 and CA-125, resulting in increased CEA concentrations to be more likely the result of true progression [32,33,34]. Persistently normalized CEA levels during treatment may therefore more accurately reflect the absence of minimal residual disease compared to Cyfra 21.1 or CA-125; indeed, patients with SeR of CEA had the longest median PFS and OS in our study. These results underscore the value of monitoring STMs over time to aid proper interpretation, potentially offering the opportunity to de-escalate treatment in patients who have clinical and radiological disease control and SeR of all STMs or SeS of Cyfra 21.1 or CA-125 and to escalate treatment to improve outcomes in patients with SeP of all STMs or SeS of CEA before clinical and/or radiological progression occurs [6,31,35].

Continuation of ICI-containing treatment in combination with local radiotherapy can improve survival for a subgroup of advanced NSCLC patients with oligometastatic disease, yet no biomarkers for diagnosing true oligoprogression are currently clinically available [36,37]. Notably, we found a significant association between the serological response patterns and the development of oligoprogression versus systemic progression, with SeS and SeP being more frequently observed in patients with oligoprogression and systemic progression, respectively. Monitoring of STM dynamics during ICI-containing treatment may therefore, together with radiological imaging, also help to more accurately identify those patients with oligoprogression that may benefit from treatment beyond progression and local radiotherapy [4,36,37].

Current guidelines do not offer any consensus recommendations on the use of longitudinal STM monitoring for the management of advanced NSCLC due to limitations in sensitivity and specificity, and the absence of decision support rules [6,25]. Our study is the first to show the value of treating STM levels as continuous variables and integrating repeated measurements to improve response classification and decision-making in clinical practice. We consider this the major strength of our study, highlighting the advantage of this approach to distinguish true progression from biological variability at the individual patient level without the need to use specific cut-off points besides the STM’s upper reference limit. Other strengths of our study include our multi-center approach, large sample size, and evaluation in a real-world setting. Models able to accurately describe the expected STM dynamics over time during treatment should be explored to determine the predictive value of the serological response patterns for assessing disease control, both in addition and compared to current clinical and radiological response assessments. Moreover, the dynamics identified in this study could potentially also be applied for other tumor cell analytes (e.g., ctDNA) facing similar challenges, as these analytes directly derive from the tumor and the dynamics are therefore expected to be similar [38,39].

Our study has several limitations. First, the retrospective design introduced inherent limitations in data collection and potential biases. Second, identification of the recurring dynamics and subclassification into the three serological response patterns was based on initial eyeballing observations and could therefore be considered subjective. For clinical practice, however, this visual classification can already be used at present and does not require complex algorithms. Last, the relatively small number of patients with a durable response and patients with STM dynamics subclassified as SeS limited our ability to draw conclusions on the added value of the serological response patterns and potential survival differences within these patient subgroups, respectively. Large multi-center prospective external validation studies are necessary to confirm generalizability of our results. Furthermore, evaluation of STM dynamics by artificial intelligence-based techniques and the addition of other tumor cell analytes may aid in subclassifying patients more objectively and be used to assess and predict treatment outcomes with higher accuracy [40].

## 5. Conclusions

Our exploratory multi-center observational cohort study shows that STM dynamics observed during the course of ICI-containing treatment could aid response classification in a real-world setting and, as such, support decision-making in clinical practice. Future studies should determine the predictive value of the serological response patterns for assessing disease control by describing the expected STM dynamics over time during treatment and exploring whether their association with survival differs between tumor- and systemic treatment-types and other tumor cell analytes to determine their generalizability and the complementarity of different tumor cell analytes for assessing and predicting treatment outcomes across multiple indications.

## Figures and Tables

**Figure 1 cancers-17-03647-f001:**
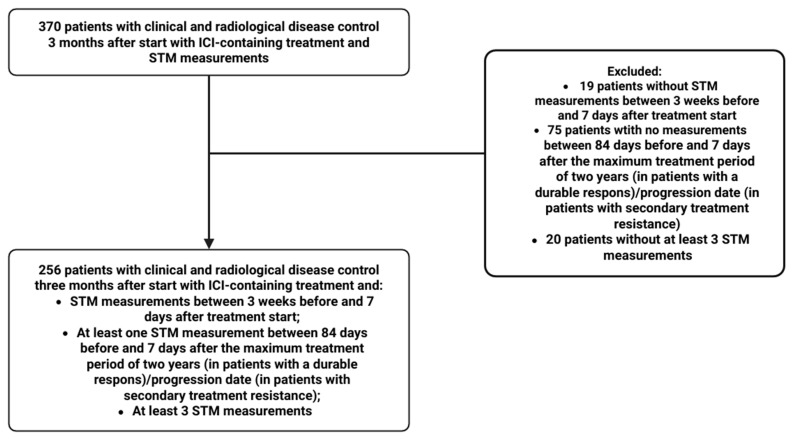
CONSORT flow diagram of patients included in the study. Abbreviations: ICI, immune checkpoint inhibitor; STM, serum tumor marker; CONSORT, Consolidated Standards of Reporting Trials.

**Figure 2 cancers-17-03647-f002:**
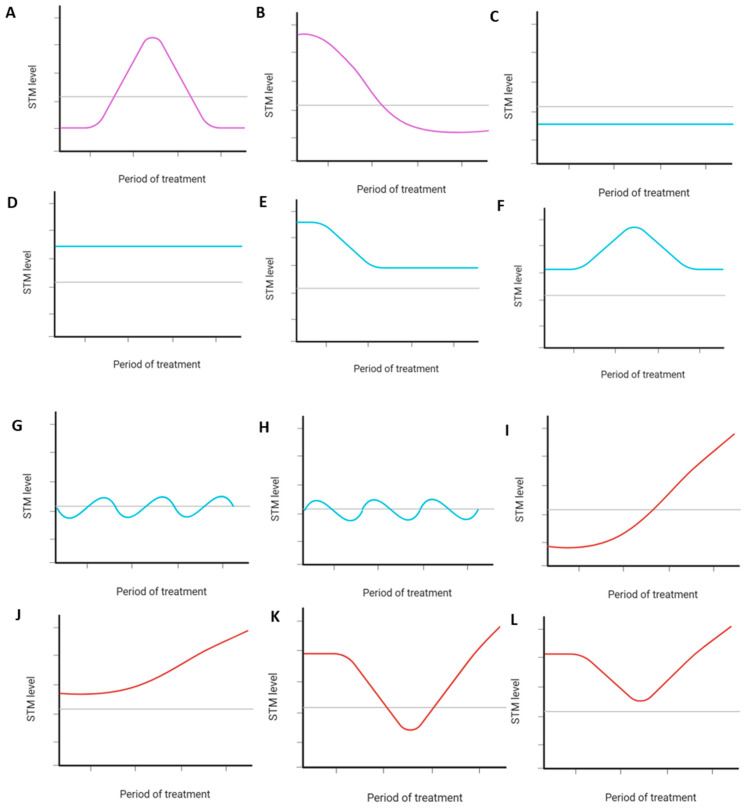
Overview of the 12 distinct recurring STM dynamics observed during treatment in advanced NSCLC patients with an initial response to ICI-containing treatment. The colored solid lines represent the distinct recurring STM dynamics observed during ICI-containing treatment. The black dotted line represents the STM’s upper reference limit. The period of treatment includes the treatment start date to seven days after the maximum treatment duration (in patients who achieve a durable response) or the date of progression (in patients who develop secondary treatment resistance). Dynamics (**A**,**B**) were subclassified as SeR, while dynamics (**I**–**L**) were subclassified as SeP. All other dynamics (**C**–**H**) were categorized as SeS. Abbreviations: STM, serum tumor marker; NSCLC, non-small cell lung cancer; ICI, immune checkpoint inhibitor; SeR, serological remission; SeS, serological stable/unknown significance; SeP, serological progression.

**Figure 3 cancers-17-03647-f003:**
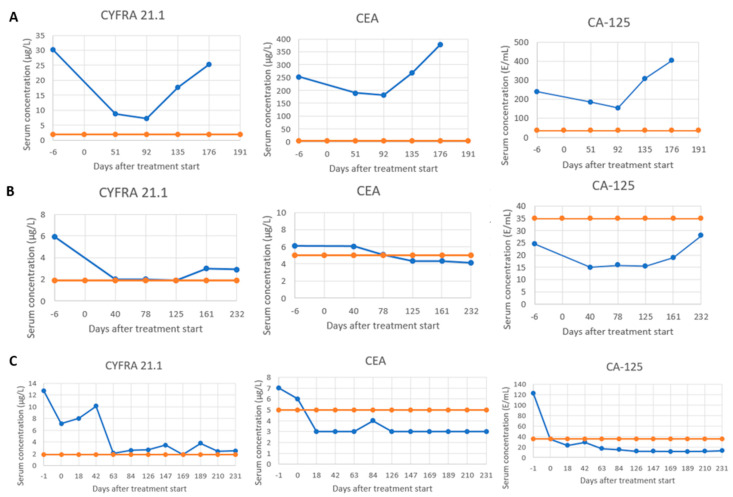
Example of STM levels visualized for three patients and their allocated individual and combined serological response classifications. The blue and orange solid lines represent the measured STM dynamics and the STM’s upper reference limit, respectively. (**A**) Patient with an individual serological response classification of SeP (dynamic L) for all STMs, resulting in a combined serological response classification of SeP. (**B**) Patient with an individual serological response classification of SeP (dynamic L), SeR (dynamic B), and SeS (dynamic C) for Cyfra 21.1, CEA, and CA-125, respectively, resulting in a combined serological response classification of SeP. (**C**) Patient with an individual serological response classification of SeS (dynamic E), SeR (dynamic B), and SeR (dynamic B) for Cyfra 21.1, CEA, and CA-125, respectively, resulting in a combined serological response classification of SeS. Abbreviations: Cyfra 21.1, cytokeratin 19 fragment antigen; CEA, carcinoembryonic antigen: CA-125, cancer antigen-125; STMs, serum tumor markers; SeP, serological progression; SeR, serological remission; SeS, serological stable/unknown significance.

**Figure 4 cancers-17-03647-f004:**
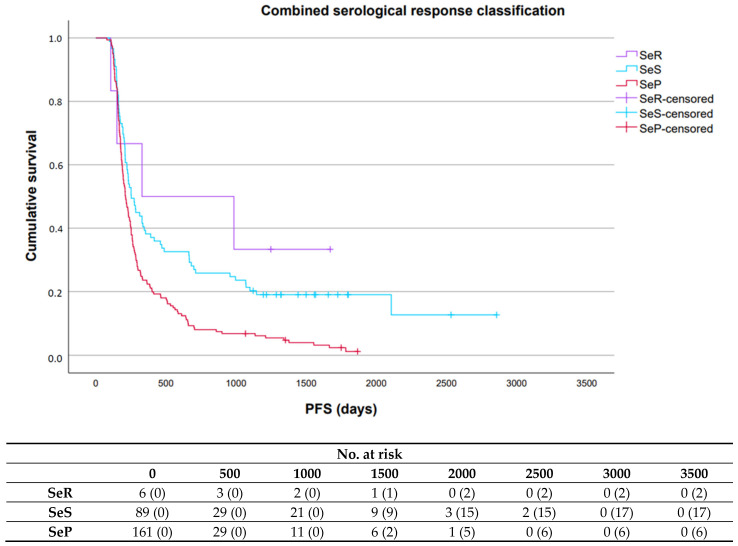
Progression-free survival associated with the three serological response patterns for the combined serological response classification. Abbreviations: SeR, serological remission; SeS, serological stable/unknown significance; SeP, serological progression; PFS, progression-free survival; No., number.

**Figure 5 cancers-17-03647-f005:**
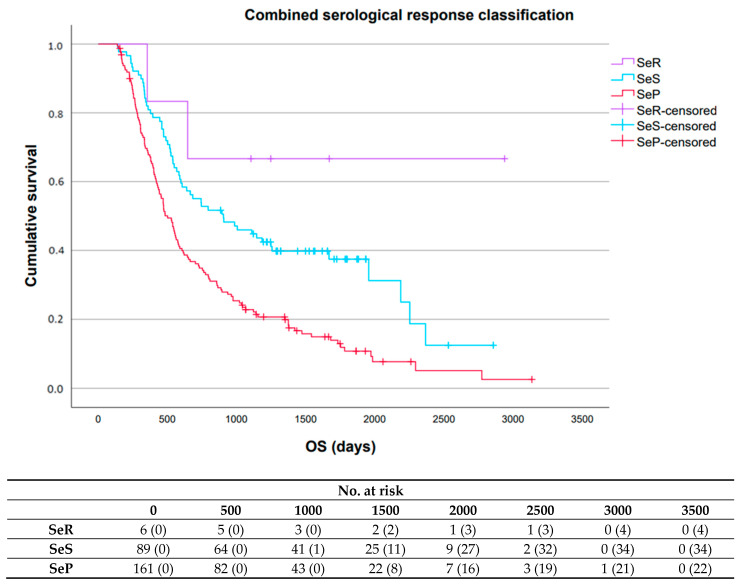
Overall survival associated with the three serological response patterns for the combined serological response classification. Abbreviations: SeR, serological remission; SeS, serological stable/unknown significance; SeP, serological progression; OS, overall survival; No., number.

**Table 1 cancers-17-03647-t001:** Patient characteristics of total patient population at baseline.

	Patients ^a^, *n* = 256 (100%)
General characteristics	
Age, median (IQR)	64.0 (56.0–71.0)
Gender (male), No. (%)	126 (49.2)
BMI, median (IQR)	24.4 (22.1–27.5)
ECOG PS (≥2), No. (%)	25 (9.8)
Ethnicity (Caucasian), No. (%)	231 (93.1)
Albumin level < 35 g/L, No. (%)	91 (37.0)
Kidney function ≤ 60 mL/min/1.73 m^2^, No. (%)	40 (15.9)
Smoking	
Smoking status, No. (%)	
Never smoker	24 (9.8)
Ex-smoker	168 (68.9)
Current smoker	52 (21.3)
Pack years, median (IQR)	30.0 (16.0–40.0)
Tumor characteristics	
Histology, No. (%)	
Adenocarcinoma	190 (75.4)
Squamous cell carcinoma	40 (15.9)
Other	22 (8.7)
PD-L1 expression, No. (%)	
Negative (<1%)	91 (42.7)
Weak positive (1–49%)	48 (22.5)
Strong positive (≥50%)	74 (34.7)
Mutation status, No. (%)	
KRAS positive	84 (36.4)
EGFR positive	29 (12.4)
BRAF positive	17 (7.0)
ALK positive	2 (0.9)
Lung cancer stage, No. (%)	
Stage ≤ III	18 (7.2)
Localization of distant metastases, No. (%)	
Brain	48 (18.8)
Bone	70 (27.3)
Liver	24 (9.4)
Adrenal gland(s)	50 (19.5)
Serum tumor markers	
Elevated baseline levels (yes), No. (%)	
Cyfra 21.1	167 (84.8%)
CEA	119 (62.0%)
CA-125	103 (59.5%)
Treatment	
Current, No. (%)	
ICI monotherapy	114 (44.5)
Nivolumab	59 (51.8)
Pembrolizumab	48 (42.1)
Atezolizumab	5 (4.4)
Other	2 (1.8)
Dual ICI therapy	1 (0.0)
Nivolumab + ipilimumab	1 (100.0)
ICI and chemotherapy	141 (55.1)
Pembrolizumab + cis-/carboplatin + paclitaxel	22 (15.6)
Pembrolizumab + cis-/carboplatin + pemetrexed	88 (62.4)
Atezolizumab + bevacizumab + carboplatin + paclitaxel	30 (21.3)
Other	1 (0.0)
Line of treatment, No. (%)	
≥Second-line	120 (46.9)
Site of inclusion	
Radboudumc (yes), No. (%)	151 (59.0)

Abbreviations: IQR, interquartile range; No., number; BMI, body mass index; ECOG PS, Eastern Cooperative Oncology Group Performance Score; PD-L1, programmed death-ligand 1; KRAS, Kirsten rat sarcoma virus; EGFR, epidermal growth factor receptor; BRAF, B-Raf proto-oncogene; ALK, anaplastic lymphoma kinase; Cyfra 21.1, cytokeratin 19 fragment antigen; CEA, carcinoembryonic antigen; CA-125, cancer antigen-125; ICI, immune checkpoint inhibitor; Radboudumc, Radboud University Medical Centre. ^a^ Missing data: BMI (*n* = 1), ethnicity (*n* = 8), albumin level (*n* = 10), kidney function (*n* = 4), smoking status (*n* = 12), pack years (*n* = 46), histology (*n* = 4), PD-L1 expression (*n* = 43), KRAS status (*n* = 25), EGFR status (*n* = 23), BRAF status (*n* = 13), ALK status (*n* = 28), lung cancer stage (*n* = 5), brain metastases (*n* = 5), bone metastases (*n* = 5), liver metastases (*n* = 5), adrenal gland(s) metastases (*n* = 5), baseline Cyfra 21.1 levels (*n* = 59), baseline CEA levels (*n* = 60), baseline CA-125 levels (*n* = 66).

## Data Availability

The datasets supporting the conclusions of this article are available in the Radboud Data Repository upon reasonable request by contacting the corresponding author (DOI: 10.34973/xs2b-v018).

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
