# Peer review of "Serological Response Patterns to Assess Treatment Outcomes in Advanced Non-Small Cell Lung Cancer: A Real-World Exploratory Multi-Center Observational Cohort Study"

_cancers, 2025, doi:10.3390/cancers17223647_

Round 1
Reviewer 1 Report
Comments and Suggestions for Authors
This is a helpful paper. Please kindly consider the issues listed below.
- The classification of serological response patterns (SeR, SeS, SeP) is based on visual (“eyeballing”) assessment by two investigators, which might introduce some subjective bias. Please address this limitation.
- Survival associations are only tested by Kaplan–Meier and log-rank analyses. No Cox regression analyses are conducted, adjusting for confounding variables such as age, sex, PD-L1 status, ECOG performance status, line of therapy, and histology, among others. This limitation hampers the interpretability of prognostic independence.
- Despite including KRAS, EGFR, and BRAF data, the analysis of molecular subtypes in relation to STM dynamics is not performed.
- Some subgroups, such as SeR = 6/256, are minimal, which leads to unstable survival estimates.
- Patients received ICI mono vs chemo-ICI across lines of therapy and access routes (trial, compassionate, routine), but survival analyses are unadjusted.
- CEA, CYFRA21-1, and CA-125 have different indices of individuality and reference change values; analyses did not model analytical CV, lot/platform effects, or apply RCV-based rules for rise/fall. This is important.
- Please improve the quality and resolution of all figures.
Author Response
- “The classification of serological response patterns (SeR, SeS, SeP) is based on visual (“eyeballing”) assessment by two investigators, which might introduce some subjective bias. Please address this limitation”
Thank you for your comment. Indeed, classification of the serological response patterns based on visual (“eyeballing”) assessment might introduce some subjective bias. We therefore included this limitation in the discussion section (line 377-379), and addressed it by suggesting that evaluation of STM dynamics by statistical techniques such as machine learning or artificial intelligence may aid in subclassifying patients more objectively in future studies (line 385-388). As our current data set is too small to correctly use these statistical techniques, however, we would like to point out that our study only served as a proof-of-concept to first determine whether STM dynamics as observed by clinicians in clinical practice can indeed be used for risk stratification. We therefore added the term “proof-of-concept” to the introduction section of the main text (line 85).
Changes to the manuscript: Introduction section, page 2, “In this proof-of-concept study, we aimed to (i) determine which STM dynamics recur during treatment in advanced NSCLC patients with disease control three months after start with ICI-containing treatment, (ii) subclassify these dynamics into three serological response patterns (e.g. serological remission (SeR), serological stable/unknown significance (SeS), and serological progression (SeP)), and (iii) explore the prognostic value of these patterns in terms of progression-free survival (PFS) and OS”.
- “Survival associations are only tested by Kaplan–Meier and log-rank analyses. No Cox regression analyses are conducted, adjusting for confounding variables such as age, sex, PD-L1 status, ECOG performance status, line of therapy, and histology, among others. This limitation hampers the interpretability of prognostic independence”
Thank you for pointing this out. We agree that Cox regression analyses improve the interpretability of prognostic independence of the identified serological response patterns. We therefore additionally performed Cox regression analyses for the individual serological response classifications and the combined serological response classification, including all patient characteristics that were described at baseline as variables in the analyses. We added this to the method section of the main text (line 180-184) and added the results of the analyses to the results section of the main text (line 302-309) and to the Supplement. Importantly, results obtained with the Cox regression analyses showed that the individual serological response classification for Cyfra 21.1 and the combined serological response classification remained significant predictors for both PFS and OS, while the individual serological response classification for CEA and CA-125 remained significant predictors for OS only.
Changes to the manuscript: Method section, page 4-5, “Statistical significance was evaluated using the log-rank test. In addition, multivariable Cox regression analyses were performed for the individual serological response classifications and the combined serological response classification, including all patient characteristics that were described at baseline as variables in the analyses”.
Changes to the manuscript: Results section, page 12, “Results obtained with the multivariable Cox regression analyses can be found in Supplemental Table S18-S21. Notably, the individual serological response classification for Cyfra 21.1 (hazard ratio (HR): 1.982 (95% CI: 1.356-2.897), p<0.001; HR: 2.654 (95% CI: 1.823-3.863), p<0.001) and the combined serological response classification (HR: 1.563 (95% CI: 1.039-2.353), p=0.03; HR: 2.726 (95% CI: 1.682-4.418), p<0.001) remained significant predictors for both PFS and OS, respectively. The individual serological response classification for CEA (HR 1.826 (95% CI: 1.290-2.583), p<0.001) and CA-125 (HR 1.774 (95% CI: 1.213-2.595), p=0.003) remained significant predictors for OS only”.
- “Despite including KRAS, EGFR, and BRAF data, the analysis of molecular subtypes in relation to STM dynamics is not performed”
Thank you for your comment. Based on results obtained with the Fisher’s exact test, we observed that patients with SeR were more frequently EGFR positive compared to patients showing SeS or SeP during treatment for CA-125 (Supplemental Table S4). For Cyfra 21.1, CEA and the combined serological response classification, no significant difference in mutation status was observed across the three serological response patterns (Supplemental Table S2-S3 and Supplemental Table S5, respectively).
- “Some subgroups, such as SeR = 6/256, are minimal, which leads to unstable survival estimates”.
Thank you for pointing this out. Indeed, we agree that some subgroups are minimal and that this leads to unstable survival estimates. We therefore included this limitation in the discussion section (line 381-384), and addressed it by underscoring that large multi-center prospective external validation studies are necessary to confirm the generalizability of our results (line 384-385).
- “Patients received ICI mono vs chemo-ICI across lines of therapy and access routes (trial, compassionate, routine), but survival analyses are unadjusted”.
Thank you for pointing this out. Based on your suggestion (e.g. comment 2), we performed Cox regression analyses for all individual serological response classifications and for the combined serological response classification, including the type of treatment, line of treatment, and site of inclusion as variables in the analyses. The results are incorporated in the results section of the main text (line 302-309) and in the Supplement as detailed in our response to comment 2.
- “CEA, CYFRA 21-1, and CA-125 have different indices of individuality and reference change values; analyses did not model analytical CV, lot/platform effects, or apply RCV-based rules for rise/fall. This is important”
Thank you for your comment. Indeed, we did not explicitly model analytical CV, lot/platform effects, or RCV-based rules for rise/fall in our analyses even though the three STMs have different indices of individuality and reference change values. As described in the method section of the main text (line 151-160), we did subclassify the identified STM dynamics into the three serological response patterns based on the STM concentration change over time within the individual patient in combination with the evaluation of the STM concentration itself with respect to the STM’s upper reference limit, thereby taking into account differences in individuality indices and reference change values across the three different STMs as shown in Figure 3 of the main text for three patients. Notably, we calculated the RCV for these three patients with the ELFM database (biologicalvariation.eu) using the given biological within-subject CV and our long-term analytical CV on our Roche system for CEA and CA-125, and based on Coskun et al. (Clin Chem Lab Med, 2021) for Cyfra 21.1. The dynamics in height of the STM considered significant by visual (“eyeballing”) assessment coincides with a significant RCV for all response classifications for each patient, except for the SeS classification for CA-125 in Figure 3B. Although a significant rise in CA-125 both visually and according to RCV was observed here, its dynamic was classified as SeS due to the low concentration of the STM below the upper reference limit. This demonstrates the concordance between visual (“eyeballing”) assessment and the RCV-based assessment, and its nuanced difference (e.g. the RCV-based assessment does not take into account the STM concentration itself with respect to the STM’s upper reference limit).
We would like to highlight that only Roche platforms were used in this study and that RCV-based rules have some limitations, amongst other that the biological variation used is obtained from healthy volunteers (and not patients) and that RCV-based rules are primarily a statistical metric based on this parameter. In our real-world cohort, all relevant and realistic analytical and biological variations are included in the data set. Therefore, it is assumed this reflects a real-world operation over a multi-year period, including any variation related to the analytical (e.g. lots, calibrations, etc.), (pre-)analytical variations, as well as within person biological variations in the obtained models. We believe our study therefore reflects a true operational performance of these measurement systems in clinical practice.
- “Please improve the quality and resolution of all figures”
Thank you for your suggestion. We improved the quality and resolution of all figures, both in the main text and in the Supplement.
Reviewer 2 Report
Comments and Suggestions for Authors
This study proposes the usage of serum tumor markers as an element for assessing prognostic outcomes for patients with non-small cell lung cancer. This retrospective study is well designed, and the limitations are well explained. Given the nature of STM being inexpensive and minimally invasive as the study states, this study provides a valuable insight for a way to track how NSLC patients will respond to treatment. However, the following minor improvements could be made:
- Methodology: Including supplemental figure Figure S1 of the flow diagram to the actual manuscript could be helpful to be more transparent about the exclusion criteria and the exact number of patients that were analyzed for the study.
- Figure 4 could be improved by matching the figure legend to the labels on the table as SeR, SeS, and SeP.
Author Response
“This study proposes the usage of serum tumor markers as an element for assessing prognostic outcomes for patients with non-small cell lung cancer. This retrospective study is well designed, and the limitations are well explained. Given the nature of STM being inexpensive and minimally invasive as the study states, this study provides a valuable insight for a way to track how NSCLC patients will respond to treatment”.
Thank you for your kind feedback.
“However, the following minor improvements could be made”:
- “Methodology: Including supplemental figure Figure S1 of the flow diagram to the actual manuscript could be helpful to be more transparent about the exclusion criteria and the exact number of patients that were analyzed for the study”
Thank you for your suggestion. We moved Supplemental Figure S1 from the Supplement to the main text.
- “Figure 4 could be improved by matching the figure legend to the labels on the table as SeR, SeS, and SeP”
Thank you for pointing this out. Indeed, we agree that Figure 4 could be improved by matching the figure legend to the labels on the table. We therefore changed the figure legend labels into the corresponding labels on the table (e.g. SeR, SeS, and SeP).